# CA-CGNet: Component-Aware Capsule Graph Neural Network for Non-Rigid Shape Correspondence

**Yuanfeng Lian ***  **and Mengqi Chen**

Beijing Key Lab of Petroleum Data Mining, Department of Computer Science and Technology,
China University of Petroleum, Beijing 102249, China
* Correspondence: lianyuanfeng@cup.edu.cn

**Abstract:** 3D non-rigid shape correspondence is significant but challenging in computer graphics, computer vision, and related fields. Although some deep neural networks have achieved encouraging results in shape correspondence, due to the complexity of the local deformation of non-rigid shapes, the ability of these networks to identify the spatial changes of objects is still insufficient. In this paper, we design a Component-aware Capsule Graph Network (CA-CGNet) to further address the features of embedding space based on the component constraints. Specifically, the dynamic clustering strategy is used to classify the features of patches produced by over-segmentation in order to further reduce noise interference. Moreover, aiming at the problem that existing routing ignores the embedding relationship between capsules, we propose a component-aware capsule graph routing to fully describe the relationship between capsules, which regards capsules as nodes in the graph network and constrains nodes through component information. Then, a knowledge distillation strategy is introduced to improve the convergence speed of the network by decreasing the parameters while maintaining accuracy. Finally, a component pair constraint is added to the functional map, and the component-based semantic loss function is proposed, which can compute isomeric in both direct and symmetric directions. The experimental results show that CA-CGNet improves by 10.21% compared with other methods, indicating the accuracy, generalization, and efficiency of our method on the FAUST, SCAPE, TOSCA, and KIDS datasets.

**Keywords:** shape correspondence; capsule network; dynamic clustering; knowledge distillation

## 1. Introduction

With the continuous development of computer 3D imaging technology, shape correspondence has become an important research direction in computer vision, which is closely related to object recognition, 3D reconstruction, image retrieval, image analysis, and other problems. The shape correspondence problem can be summarized as identifying homologous points of two or more shapes, where a large number of variables are needed to define a dense map for a non-rigid object due to its complex changes such as deformation, distortion, and extension. Although this problem has made some breakthroughs [1–4], finding dense shape correspondence is still very challenging.

Traditional non-rigid shape correspondence methods are generally established by measuring the similarity of point descriptors between two shapes, such as spectral geometry [5] or diffusion distance [6]. However, these methods rely heavily on initialization, which falls easily into local optimization with slow matching speed and low matching accuracy. Faced with large amounts of data and dense correspondence, scholars have further proposed to obtain the features of each point on the manifold surface through neural networks [7–9]. Unfortunately, these techniques directly regress the dense relationship of all input points, severely over-parameterizing the deformation and leading to poor generalization. Recent efforts mainly focus on functional map frameworks [10–13] that converts point-to-point correspondence between models into a linear transformation in function space, mapping

a point on the source model to multiple points on the target model to solve ambiguity problems. Despite significant progress in this area, these methods ignored the hierarchical geometric relations of the associated semantic parts. When the differences between the two input models are large, the extracted non-rigid shape descriptors lack sensitivity to specific information such as orientation and position, leading to unreliable correspondence results.

Based on the above limitation, we propose a novel capsule graph neural network based on component constraint for the 3D non-rigid shape correspondence, called CA-CGNet, to further capture descriptors with powerful representation abilities. The entire framework of our method is given in Figure 1. The over-segmentation method is used to segment the mesh into several patches, which are clustered to form components using a dynamic clustering algorithm. In order to further improve the quality of feature acquisition, we design a novel graph routing algorithm to learn the relationship between the capsules in the same layer through a component-aware graph neural network (CA-GNN). Then, the knowledge distillation strategy is used to improve the convergence speed of the network by reducing the parameters while maintaining accuracy. Finally, a component pair constraint is added to the functional map with a component-based semantic loss function, which makes the shape correspondence more accurate and improves the robustness of the network.

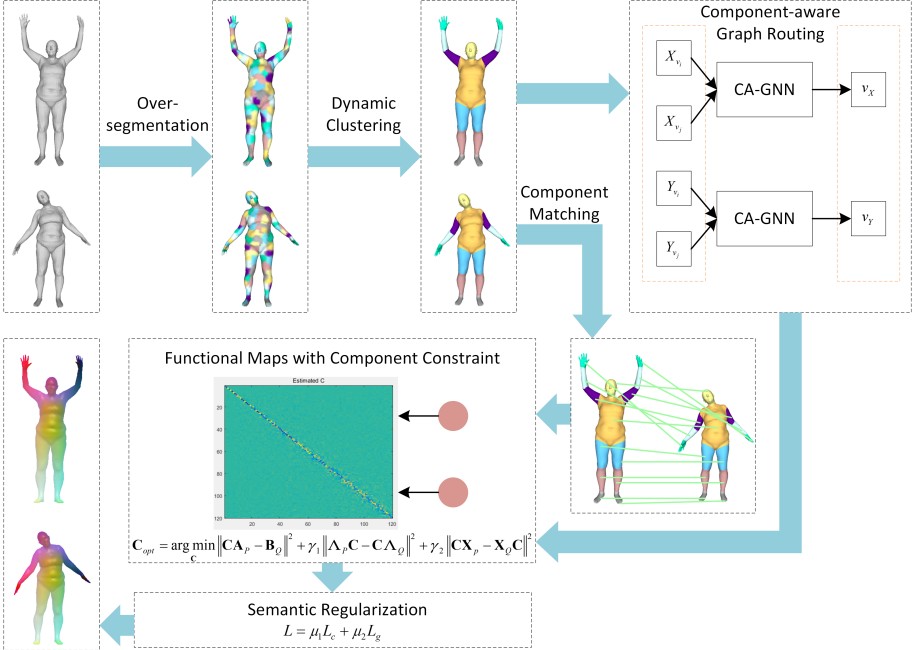

**Figure 1.** The pipeline of component-aware capsule graph neural network framework.

Our main contributions can be summarized as follows:

- We design a novel network structure called CA-CGNet, which enhances the expressive ability of the network to represent object spatial pose and orientation changes by adding local mesh pose details to achieve high-precision shape correspondence.
- To reduce the interference of noisy patch assignments, we propose a dynamic clustering algorithm to cluster the over-segmented meshes dynamically according to feature similarity to form components. By adding a component constraint to the functional maps and integrating component-based semantic constraint loss into the regularization term, the accuracy of shape correspondence is further improved.
- The component-aware graph routing treats capsules as nodes in a graph neural network by adding component constraints to obtain more accurate relationships between capsules. In addition, the knowledge distillation strategy is used to reduce the number of parameters while maintaining network performance.
- Experiments on four challenging datasets show qualitatively and quantitatively that the CA-CGNet has stronger robustness and better generalization. The ablation study

demonstrates that component pair constraint, component-aware graph routing, and knowledge distillation strategy have a great improvement in network performance.

## 2. Related Work

### 2.1. Non-Rigid Shape Correspondence

Various work has been done in the direction of shape correspondence. Before deep learning methods were commonly applied to point descriptor learning, many scholars conducted in-depth research on handcrafted feature descriptors for characterizing the geometric information of shapes. Tombari et al. [14] proposed the signature of histograms of orientations (SHOT) descriptor with rotation and translation invariance by counting the topological features adjacent to points in the local coordinate system and saving them in the histogram. Average geodesic distance (AGD) [15] was the attitude insensitive descriptor, which could effectively measure the overall topological characteristics of multi-resolution models. Sun et al. [16] extracted all key points in different proportions and combined them with the thermal diffusion equation to develop a heat kernel signature (HKS) with multi-scale characteristics. In [17], the wave kernel signature (WKS) used the Schrodinger equation to acquire the evolution process of particles on the shape surface to obtain the features of the vertex on the object surface. To explore posture more effectively, Zuffi et al. [18] proposed to use graphical nodes corresponding to body parts so that these parts could be independently translated and rotated in 3D to represent different body shapes and to capture pose-dependent shape variations. This method was similar to our approach but only applied to human models.

Recently, researchers have tried to use deep learning methods to further process shape descriptors to obtain high-quality geometric features of 3D non-rigid surfaces [19,20]. Cyclic-FM [21] was based on a cyclic mapping between metric spaces for self-supervised dense correspondence mapping between non-isometric shapes. Marin et al. [22] extracted stable landmarks over human bodies, relying entirely on the geometric properties in the spectral domain. The MGCN [23] advanced a multi-scale graph convolutional network, which transformed wavelet energy decomposition signature (WEDS) to a more discriminative descriptor. Amor et al. [24] made use of deep residual neural networks to calculate minimizing paths between deformations and, thus, between shapes for the alignment of 3D shapes under complex topology-preserving transformations. Although these methods have achieved good results, they did not make full use of local pose information to maintain the geometrical relationship of the manifold. In addition, when the model resolution decreases, the discriminative power of the network decreases significantly.

### 2.2. Capsule Graph Neural Network

Capsule network was an improved strategy for CNN proposed by Hinton et al. [25], which used vectors to represent the mutual relations between features. We find the outcomes and limitations of several approaches based on a capsule graph neural network in Table 1. The basis of the capsule network was the routing process, which passed the input vector to the upper capsule through the protocol. To effectively preserve the variability and correlation between features during the routing process, capsule graph neural networks [26] extracted node features in the form of capsules to capture high-quality node embeddings at the graph level. In [27], Caps-GNN was adopted to learn graph properties for encoding underlying characteristics. CapsGNNEM [28] utilized EM routing to obtain graph attributes from the node features extracted by the graph neural network. Some other research [29,30] represented nodes as a group of node-level capsules, jointly learning node embedding and extracting salient features of corresponding nodes through heterogeneous factors of the capsules. Later in [31], the method was proposed to model the relationship between concept capsules of the same layer through a graph network with an external storage matrix. In our work, we propose a novel routing algorithm that leverages component constraints to transform primary capsules into high-quality graph embedding, which better maintains the local posture of the model.

**Table 1.** A comparative study for Capsule Graph Neural Network in open literature.

| Reference | Routing Methods | Outcomes | Limitations |
|---|---|---|---|
| CapsGNN [26] (2018) | Dynamic routing | Higher accuracy rate compared to traditional methods | Irrelevant messages from multi-hop neighborhoods has not been restrained |
| Caps-GNN [27] (2020) | Dynamic routing | Higher inference in personalized preference | External knowledge has not been considered |
| HGCN [30] (2021) | Nonlinear function | More effectively capturing the heterogeneous factors under each node. | The over-smoothing issue over graph is ignored |
| CapsGNNEM [28] (2021) | EM routing | Higher graph classification compared to standard methods | Structural information of the graph has not been considered |
| NCGNN [29] (2022) | Dynamic routing | Adaptively identifying a subset of crucial node-level capsules | Unable to preserve structure information of lower-level parts |
| Caps-HAGKT [31] (2022) | Capsule routing | Extracting the latent knowledge structure between levels | Automatic modeling of the complex knowledge structure of the knowledge capsule at same layer is insufficient |
| Ours | Component-aware graph routing | Using component constraints to solve problem of model posture details and low resolution | Performance can be improved with optimizing selection of the number of components |

*2.3. Correspondence with Functional Maps*

Different from point-to-point correspondence, the concept of the functional map was introduced in [32], which represented the mapping between shapes as a small matrix and transmitted information by encoding the relationship between the basic functions of shapes. Inspired by that, some scholars have proposed several learning methods for optimizing functional maps [10,11,33]. With the development of deep learning, a neural network was used in the functional map framework to generate more accurate correspondence. Litany et al. [12] proposed a deep neural network architecture named FMNet by providing a dense corresponding representation of linear operators through paradigm transformation and a structured prediction model. SURFMNet introduced in [13] was to use pure geometric scalars to directly establish correspondence in the 3D shape set without any prior information. In [34], Deep Geometric Functional Maps was a feature extraction network based on a functional map representation that learned directly from the shape geometry with a new regularized map extraction layer and loss to enforce the structural properties. Donati et al. [35] generalized the functional map framework to the conformal maps between tangent vector fields, which reflected the complex structure of the surface to maintain orientation-aware results but naturally restricted to differentials of conformal mappings. The key difference between the above approach and our proposed approach is that we embed component constraints in the functional maps framework, focusing on the local area feature information of the mesh.

**3. Proposed Method**

In this section, we present a component-aware capsule graph neural network for shape correspondence. In Section 3.1, we introduce the whole network architecture of CA-CGNet. Section 3.2 illustrates the process of turning input shapes into component pairs. The component-aware graph routing is given in Section 3.3. We introduce the functional map of component pair constraints in Section 3.4 and the semantic regularization term in Section 3.5.

### 3.1. CA-CGNet

In this work, we design a deep neural network named the "Component-aware Capsule Graph Neural Network" (CA-CGNet) for shape correspondence based on the functional maps framework. We believe that the capsule graph network allows a powerful understanding of the mesh's positions and directions features. Since existing routing algorithms do not fully consider the potential features between capsules, a component-aware graph capsule network is proposed to further extract deeper semantic information by building graph relationships between capsules. Then, in order to get more precise shape correspondence, the functional map with component constraint and the loss function with semantic constraint as the regularization terms are designed. The basic pipeline can be described by the following steps: firstly, the mesh is over-segmented to form several patches, and the patches with similar features are aggregated together to form components through a dynamic clustering algorithm. Secondly, in order to fully learn the relationships between capsules, we consider capsules as nodes of the graph neural network and further enhance the capsules through a component-aware graph routing algorithm with a knowledge distillation strategy to increase the convergence speed by decreasing the number of parameters while maintaining network accuracy. Finally, the component-constrained functional map is represented with Equation (11), and the loss function with semantic constraint is derived from Equation (17). The entire network architecture is given in Figure 2 in detail.

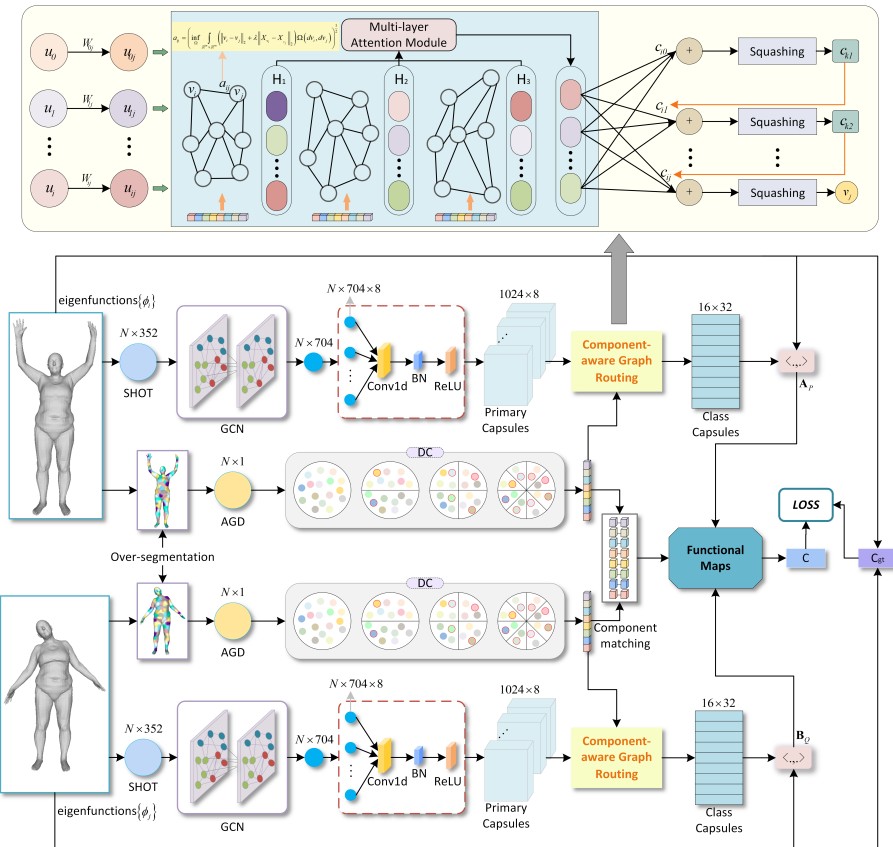

**Figure 2.** The architecture of our proposed network CA-CGNet. For the mesh component, dynamic clustering (DC) is applied to form the patches generated by over-segmentation. To further improve the quality of feature acquisition, the component-aware graph routing algorithm is constructed to learn the relationship between the capsules in the same layer through a component-aware graph neural network (CA-GNN). The component constrained functional map is represented by Equation (11) and the component-based semantic loss is derived from Equation (17).

### 3.2. Component Extraction

A 3D model usually contains a large number of faces, which can generate a huge amount of computation if the entire mesh model is segmented directly. Therefore, the mesh sub-module is constructed using the method proposed by Golovinskiy et al. [36] to over-segment the 3D mesh into several small patches. Since the number of vertices contained in each patch is inconsistent, Principal Components Analysis (PCA) is used to reduce the dimensionality to make the number of vertices contained in each patch equal. After that, the Average Geodesic Distance (AGD) feature of each patch is calculated, and the sub-model is aggregated into components with dynamic clustering. Finally, the improved Hausdorff distance is used for component matching to extract the component pairs.

#### 3.2.1. Dynamic Clustering

Mesh segmentation is equivalent to the clustering task of a 3D surface. To further reduce noise interference, we use dynamic clustering strategy to perform more accurate part-to-whole classification of the acquired features. In the DC part of Figure 2, small circles with different colors represent different types of features, and the circles with red borders are used as the initial feature of clustering. The features are clustered according to the correlation, and the highly correlated features are clustered together. Generally speaking, dynamic clustering includes three steps, namely, calculating the feature correlation matrix, selecting the set of features with the largest difference as the initial feature, and clustering the remaining features.

In a 3D mesh, the properties of vertices are represented by their feature matrices. By calculating the Manhattan distance between the feature matrices of different vertices, we can obtain the degree of correlation between these features. Specifically, the degree of correlation $\mathbf{R}$ between feature $X$ and $Y$ is calculated with

$$\mathbf{R}(X, Y) = \sigma\left(\sum_{i=1}^{n} |x_i - y_i|\right) \tag{1}$$

where $X = (x_1, x_2, \cdots, x_n)$ and $Y = (y_1, y_2, \cdots, y_n)$ are both $n$-dimensional vectors, $\sigma$ represents the sigmoid activation function.

In the correlation matrix $\mathbf{R} \in \mathbb{R}^{N \times N}$, the degree of correlation between features decreases with the increase of the coefficient. By choosing the coefficient with the largest value in $\mathbf{R}$, we can obtain the row coordinate $r_1$ and column coordinate $r_2$ of this value, which represent the sequence numbers of the two groups of features with the farthest correlation. Then, these two features are added to set $G$ and respectively used as the initial features for each cluster.

The value of the $i$-th row in matrix $\mathbf{R}$ represents the degree of correlation between the $i$-th type feature and other types of features. The cluster of the remaining features is expressed as follows:

$$r_i = \arg\min_{r_j \in G}\left(\mathbf{R}_{r_j, \varepsilon}\right), \varepsilon \neq r_j \tag{2}$$

where $r_j$ represents the sequence number of the initial two types of features, and $\varepsilon$ represents the sequence number of the remaining features. Thus, these features can dynamically be divided into two clusters. Repeating the above steps on each cluster, we will repartition each cluster into two new sets of feature clusters. After executing this process three times, the over-segmented patches are clustered into several types.

#### 3.2.2. Component Matching

The Hausdorff distance [37] reflects the gap of two sets by measuring the distance between two nonempty subsets. Using Hausdorff distance for model matching can simplify the calculation, but it is frequently interfered with by noise. In order to build the

corresponding relationship between the components of the two models, this paper adopts the improved Hausdorff distance to match the components.

For two sets $A = \{a_1, a_2, \cdots, a_i\}$ and $B = \{b_1, b_2, \cdots, b_j\}$, the bidirectional Hausdorff distance between these two sets is defined as follows:

$$H(A, B) = \max\{h(A, B), h(B, A)\} \tag{3}$$

where $h(A, B)$ denotes the unidirectional Hausdorff distance from set $A$ to set $B$, $h(B, A)$ denotes the unidirectional Hausdorff distance from set $B$ to set $A$. The improved unidirectional Hausdorff distance is written as:

$$h(A, B) = \frac{1}{N} \sum_{a \in A} \left\{ \min_{b \in B} \|a - b\| \right\} \tag{4}$$

where $N$ represents the number of elements in set $A$, $\| \cdot \|$ represents the distance between two elements in different sets.

We first select component $A$ in the manifold $M$ and calculate its Hausdorff distance to each component in the manifold $N$ to obtain the distance set $D = \{d_1, d_2, \cdots, d_k\}$. Then we choose the minimum distance $d_{\min}$ in the set to form a component pair and delete the two components from the original set. We repeat the steps for the remaining components in turn to construct a pair of components.

### 3.3. Component-Aware Graph Routing

Considering the CapsNet can capture the part-whole relationship to cope with ambiguity [25], we introduce the capsule network to enhance feature expression ability. The features extracted by GNN are replicated in eight copies, the primary capsules are obtained after Conv1d+BN+ReLU, and the pose features of the mesh are further extracted to compose the latent capsules. Among them, routing is the basis of the capsule network, which transmits the information from the previous capsule layer to the next one through the protocol. To further improve the quality of feature acquisition, we propose a component-aware graph routing, which learns the relationship between capsules in the same layer through CA-GNN.

#### 3.3.1. Multi-Layer Attention Graph Routing

We extract features from different CA-GNN layers and fuse the capsule features through a multi-layer attention mechanism. The capsule is regarded as each node in the graph, and the adjacency matrix **A** is used to represent the relationship between nodes in CA-GNN, which can be expressed as:

$$\tilde{\mathbf{A}} = \mathbf{D}^{-\frac{1}{2}} \mathbf{A} \mathbf{D}^{-\frac{1}{2}} + \mathbf{E} \tag{5}$$

where $\tilde{\mathbf{A}}$ is the normalized $N \times N$ adjacency matrix, $N$ is the number of capsules, **D** is the degree matrix, and **E** is the identity matrix.

The capsule vectors can be regarded as the probability of certain attributes, which is used to measure the semantic similarity of capsules. Since the capsule vector is an implicit expression, it is difficult to directly calculate the relationship between capsules. Thus, we used Wasserstein Distance to measure the distance between the distribution of capsule properties. The Component-aware Wasserstein Distance is defined as:

$$W(V_i, V_j) = \left( \inf_{\Omega} \int_{\mathbb{R}^m \times \mathbb{R}^m} \left( \|v_i - v_j\|_2 + \lambda \left\| X_{v_i} - X_{v_j} \right\|_2 \right) \Omega(dv_i, dv_j) \right)^{\frac{1}{2}} \tag{6}$$

where $X_{v_i}$ and $X_{v_j}$ are the component category to which the point belongs, $\Omega$ is an element of set $\Pi(V_i, V_j)$, and $\lambda$ is the coefficient and is set to be 0.005. Let $a_{ij} = W(V_i, V_j)$, we can convert component-aware Wasserstein Distance to adjacency between capsules inside. In

this case, $a_{ij}$ may be negative, so the adjacency matrix **A** needs to be normalized, which is denoted as

$$\tilde{\mathbf{A}} = \frac{e^{\mathbf{A}_i}}{\sum\limits_j e^{\mathbf{A}_i}} + \mathbf{E} \tag{7}$$

Considering that the contribution of different levels of graph embedding is inconsistent, we introduce an attention mechanism to combine these different levels of graph embedding to get the final output result as shown in Figure 3. The procedure of the multi-layer attention module can be written as:

$$w_j = \sigma\left\{ \left(H_1^{GAP}, H_1^{GMP}\right) \oplus \left(H_2^{GAP}, H_2^{GMP}\right) \oplus \left(H_3^{GAP}, H_3^{GMP}\right) \right\} \tag{8}$$

where $\sigma(\cdot)$ represents the softmax function, and $\oplus$ represents the feature concat.

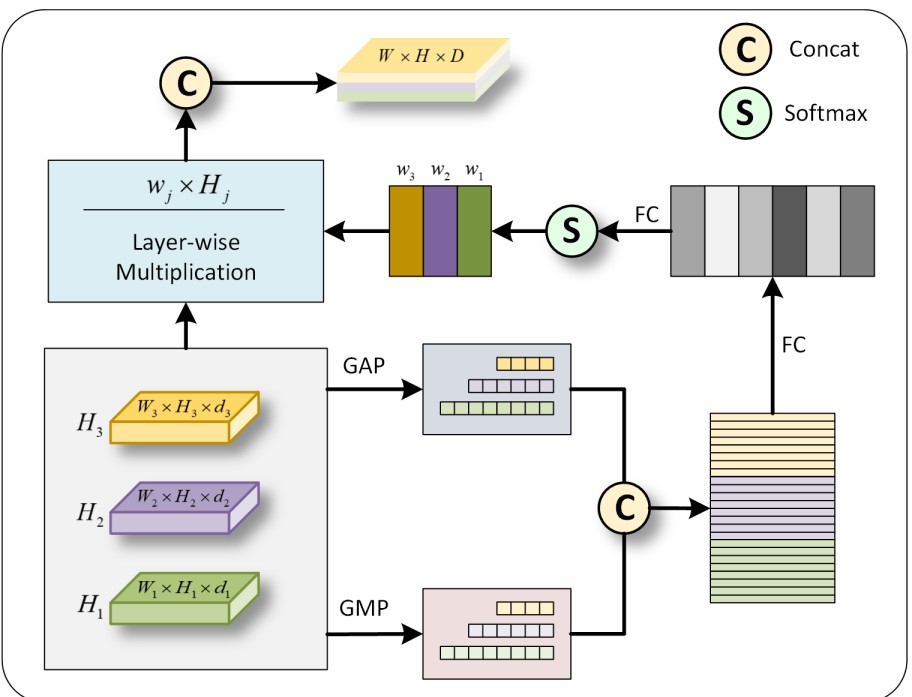

**Figure 3.** The Multi-layer Attention Module. GAP means global average pooling, while GMP refers to global maximum pooling. FC is the fully connected layer. $w_j$ means the weight for each layer.

Each capsule is regarded as a center node, and the relationship between capsules is measured by distance with the performed normalization. Then, the multi-layer attention mechanism is introduced to transform the low-level capsule into the high-level capsule using component graph routing. Algorithm 1 gives the complete algorithm of Component-aware Graph Routing.

---

**Algorithm 1:** The Algorithm of the Component-aware Graph Routing

---

　　**Input:** The low-level capsules $u_i$ and the weight matrix $\mathbf{W}_{ij}$
　　**Output:** The high-level capsules $v_j$

**1** initialize routing coefficients $b_{ij} \leftarrow 0$
**2 begin**
**3** 　*// each low-level capsule i*
**4** 　$u_{j|i} = u_i \mathbf{W}_{ij}$
**5** 　Calculate the adjacency matrix $\mathbf{A}$ for all low-level capsules
**6** 　Normalize $\mathbf{A}$ with Equation (7)
**7** 　*// each low-level capsule i and high-level capsule j*
**8** 　$g_j = u_{j|i} \tilde{\mathbf{A}} \mathbf{W}_{ij}$
**9** 　Calculate the attention score $w_j$ with Equation (8)
**10** 　**for** *n in routing interation* **do**
**11** 　　*// each low-level capsule i*
**12** 　　$c_{ij} = \frac{1}{1+e^{-b_{ij}}}$
**13** 　　*// each high-level capsule j*
**14** 　　$s_j = \sum_i g_j w_j c_{ij}$
**15** 　　$v_j = squash(s_j)$
**16** 　　$b_j = b_j + u_{j|i} v_j$
**17** 　**end**
**18** 　**return** $v_j$
**19 end**

---

### 3.3.2. Knowledge Distillation Strategy

We introduce a knowledge distillation strategy into component-aware graph routing, which can transfer knowledge from a complex deep teacher network to a simple student model. Since the node features and the associations between nodes extracted by graph neural networks play an important role in embedding features, we capture the geometric information of the latent space by extracting the features of the CA-GNN layer and using the graph information extracted from the teacher network to train the student network. Therefore, the student can be further guided by matching the node structure extracted by itself and the node structure embedded by the teacher. As shown in Figure 4, the original component-aware graph routing is defined as the teacher network, and the pruned component-aware graph routing is defined as the student network. The teacher network deploys six CA-GNN modules after the primary capsule to extract features with more semantic information. In contrast, the student network only contains three CA-GNN modules, which greatly reduces the parameters to improve the convergence speed of the network.

In order to measure local detail dependence in low dimensional space, we conduct inter-layer knowledge distillation between CA-GNN layers. The feature vectors of teacher network $I_T$ are obtained through two CA-GNN layers, and the feature vectors of student network $I_S$ are obtained through one CA-GNN layer. Then, the loss of inter-layer knowledge distillation is defined as:

$$L_{inter} = \sum \log(I_S) \log \frac{\log(I_S)}{I_T} \tag{9}$$

After the Multi-layer attention module integrates the features of each CA-GNN layer, the latent capsule containing global information is obtained. Let $O_T$ and $O_S$ be the feature

vectors of the teacher network and student network, the output layer knowledge distillation loss over all the capsules is as follows:

$$L_{out} = \frac{1}{N} \sum_{k=1}^{N} \sum_{j:(i,j)\in\varepsilon} O_{S_{ij}} \log\left(\frac{O_{S_{ij}}}{O_{T_{ij}}}\right) \tag{10}$$

where $\varepsilon$ represents edges in a graph network, $k$ donates the number of capsules.

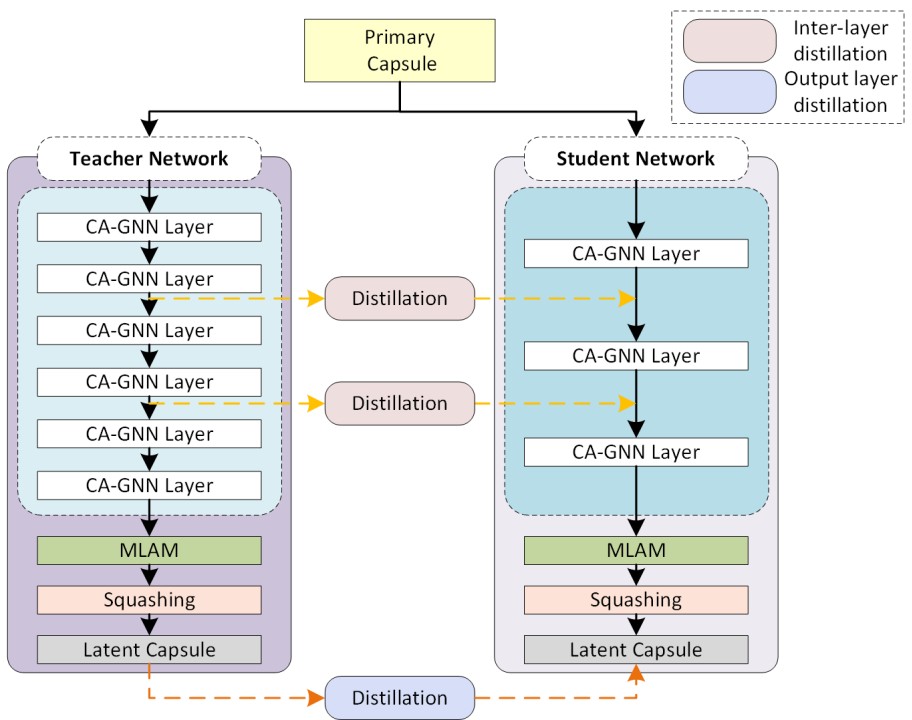

**Figure 4.** Our knowledge distillation strategy in Component-aware Graph Routing. CA-GNN means the Component-aware Graph Neural Network. $F_T$ and $F_S$ are the features of the teacher network and of the student network, respectively. MLAM is the Multi-layer Attention Module.

### 3.4. Functional Maps with Component Constraint

We propose a component constrained functional maps, which provides a local–global commutative preservation for shape correspondence between manifold $P$ and $Q$. Despite traditional functional maps providing the simplicity and efficiency of shape correspondence, its estimation pipeline solely enforces the commutativity of global operators. This method is to search matching points across the whole manifold, which leads to a large extent to some inaccurate matching. To alleviate this issue, we propose an available segmentation of shapes for relation constraints. Then, the optimal functional map **C** in the least square sense can be expressed as:

$$\mathbf{C}_{opt} = \arg\min_{\mathbf{C}} \left\|\mathbf{C}\mathbf{A}_P - \mathbf{B}_Q\right\|^2 + \gamma_1 \left\|\Lambda_P\mathbf{C} - \mathbf{C}\Lambda_Q\right\|^2 + \gamma_2 \left\|\mathbf{C}\mathbf{X}_P - \mathbf{X}_Q\mathbf{C}\right\|^2 \tag{11}$$

where $\mathbf{A}_P$ and $\mathbf{B}_Q$ are the basis matrix consisting of the basis function coefficients of the source model $P$ and the target model $Q$, $\Lambda_P$ and $\Lambda_Q$ are the diagonal matrix of Laplacian-iBeltrami eigenvalues, $\mathbf{X}_P$ and $\mathbf{X}_Q$ are the Hausdorff distance matrix, and $\gamma_1$ and $\gamma_2$ are the weights.

The first term of Equation (11) calculates the deviation of the point-wise descriptor coefficient matrix based on SHOT descriptors of manifold $P$ and $Q$. For a connected smooth compact manifold $P$ with $K$ vertices, an index function $f : P \to \mathbb{R}$ can be constructed by calculating the average value of the point and its neighborhood coordinates, which can be expressed as:

$$f(x) = \sum_i m_i \phi_i \tag{12}$$

where $m_i = \langle f, \phi_i \rangle_P$, $\{\phi_i | i = 0, 1, \ldots, k-1\}$ is the set of the eigenfunctions for the orthogonal basis of $f$, and f is the corresponding discretization of the smooth function $f$. Let $F : f(P) \to f(Q)$ be a linear operator to calculate correspondence between manifolds $P$ and $Q$. By computing a set of descriptor functions $\{\phi_i\}_{i \geq 0}$ and $\{\phi_j\}_{j \geq 0}$ on shapes $P$ and $Q$, the function $F$ can be represented as:

$$F(f) = \sum_j \sum_i m_i c_{ji} \phi_j \tag{13}$$

where $c_{ji} = \langle F(\phi_i), \phi_j \rangle_N$, and then admit a matrix representation $\mathbf{C} = (c_{ji})$. To determine the matrix $\mathbf{C}$, we truncate the eigenfunctions after the first $d$ coefficients to significantly approximate matrix $\mathbf{C}$. Assuming to be given a set of coefficients $z_i = \langle f, \phi_i \rangle$ and $g_j = \langle s, \phi_j \rangle$, the matrices $\mathbf{A}_P$ and $\mathbf{B}_Q$ can be stored by these Fourier coefficients.

The second term of Equation (11) is a regularization item that constrains the descriptor by reinforcing its overall structure properties. With the Laplacian matrix $\mathbf{L}$, the set of eigenvalues $\lambda_i$ are then defined as:

$$\lambda_i = \mathbf{L}\mathbf{M}^{-1} \tag{14}$$

where $\mathbf{M}$ is the $K \times K$ diagonal mass matrix. Thus, the first $k$ eigenvalues of Laplacian–Beltrami operator on model $P$ and $Q$ compose the diagonal matrices $\Lambda_P = diag(\lambda_i)_P$ and $\Lambda_Q = diag(\lambda_j)_Q$.

The third term of Equation (11) penalizes the local commutativity through the component constraints. Each manifold is segmented into several types of components, and Hausdorff distance is used to construct component pairs. Hausdorff distance is a measure of the similarity between two sets of points, so we can calculate the difference between components. The distance matrix $\mathbf{X}_P$ and $\mathbf{X}_Q$ can be expressed as:

$$\begin{cases} \mathbf{X}_P = h(C_P, C_Q) \\ \mathbf{X}_Q = h(C_Q, C_P) \end{cases} \tag{15}$$

where $C_P$ and $C_Q$ are the features of component pairs. Thus, each set of component pairs can be used to impose local constraints on functional maps.

### 3.5. Semantic Regularization

In order to enhance the robustness of function mapping, we transfer the component network discussed in Section 3.2 to the feature space and further develop the semantic regularization term. By penalizing the difference between the weights of each node on each component pair and the weights of corresponding nodes to regularize the classification model, we define a semantically constrained loss function to preserve the classification in the embedded feature space, which is formulated as:

$$L_c = \frac{1}{n} \sum_j^n \sqrt{\sum_i |x_{P_i} - x_{Q_i}|^2} \tag{16}$$

We integrate this semantic constrained loss function into the final loss function, then the overall spectral loss function $L$ of the corresponding shape is defined as follows:

$$L = \mu_1 L_c + \mu_2 L_g \tag{17}$$

where $\mu_1$ and $\mu_2$ are the coefficient, and $L_g$ denotes the difference between the mapping matrix **C** and the corresponding ground-truth matrix [38]. In our experiments, $\mu_1$ and $\mu_2$ are both set to be 1. By adding the semantic regularization term, the spectral loss function of shape corresponding learning network is minimized; thus, the constraints on the corresponding functions are increased during the training process.

## 4. Experiments and Evaluation

In this section, we verify the performance of 3D non-rigid shape correspondence by comparing CA-CGNet with some classical methods. This experiment runs on GeForce GTX 1080Ti CPU with 32GB memory. ADAM optimization algorithm with parameter set as $\beta_1 = 0.9$, $\beta_2 = 0.999$, $\varepsilon = 10^{-8}$ is used, and the learning rate is changed by polynomial decay strategy, where the initial learning rate is $10^{-3}$, the end learning rate is $10^{-5}$, and the batch size is 1.

### 4.1. Dataset

Four types of non-rigid 3D datasets, namely FAUST, SCAPE, TOSCA, and KIDS, are used for experiments. In the experiment, the first 80 meshes in the original FAUST dataset are trained, and the last 20 meshes are used to test the model. And its vertices are sampled to 4096 vertices to verify the robustness of the experiment. For other datasets, the vertices are sampled to 4096 to facilitate comparison experiments.

FAUST dataset has models with the same triangulation and resolution, and each model contains 6890 vertices. There are 100 models in the data set, and every 10 models is a group representing different poses of the same research object. The shapes in the dataset have strong non-isometric deformation, and the ground truth correspondence of vertices between all shapes is known.

SCAPE dataset is a data-driven approach to building a human shape model across the shape and pose variations, using a static scan and a labeled motion capture sequence of humans to artificially generate high-quality animated surface models of moving humans with realistic muscle deformations.

TOSCA dataset consists of many different types of 3D graphics, with significant differences between each set of models. Shapes in the same group are compatible, meaning that all shapes have the same mesh resolution and connectivity.

KIDS dataset is a synthetic non-isometric dataset containing two sets of human models with different poses, which have the same number of vertices and contain ground truth.

### 4.2. Component Matching

For each 3D mesh, over-segmentation is used to divide it into many different regions, and then dynamic clustering is used to merge regions to form components. The selection of the number of clusters $k$ not only affects the amount of computation but also affects the subsequent matching effect. Therefore, we use different cluster numbers with $k = 4, 5, 6, 7, 8, 9, 10$ for experimental tests. Table 2 shows the identification errors of experiments on human and non-human models with a different number of clusters. Through the experiment, it can be found that the number of clusters with $k = 8$ can achieve the minimum identification error on the human model and centaur model, and the clustering amount of $k = 5$ can achieve the minimum identification error on other non-human models. Therefore, in the subsequent experiments, we adopted the dynamic classification strategy with $k = 8$ on the human model and centaur model and $k = 5$ on other non-human models.

**Table 2.** Identification error with a different number of clusters on four datasets. Error is measured as the average Hausdorff distance between the source model and the target model (cm).

| Clustering Number | FAUST | SCAPE | KIDS | TOSCA | | | | | | | |
|---|---|---|---|---|---|---|---|---|---|---|---|
| | | | | David | Michael | Victoria | Cat | Centaur | Dog | Horse | Wolf |
| 4 | 0.0659 | 0.0872 | 0.0814 | 0.0652 | 0.0796 | 0.0684 | 0.0369 | 0.0427 | 0.0358 | 0.0374 | 0.0335 |
| 5 | 0.0586 | 0.0697 | 0.0705 | 0.0613 | 0.0686 | 0.0529 | 0.0254 | 0.0291 | 0.0216 | 0.0195 | 0.0208 |
| 6 | 0.0403 | 0.0574 | 0.0592 | 0.0486 | 0.0473 | 0.0351 | 0.0263 | 0.0314 | 0.0228 | 0.0199 | 0.0219 |
| 7 | 0.0258 | 0.0261 | 0.0243 | 0.0186 | 0.0195 | 0.0217 | 0.0269 | 0.0316 | 0.0231 | 0.0207 | 0.0231 |
| 8 | 0.0193 | 0.0201 | 0.0189 | 0.0174 | 0.0181 | 0.0203 | 0.0271 | 0.0320 | 0.0235 | 0.0214 | 0.0233 |
| 9 | 0.0206 | 0.0211 | 0.0193 | 0.0176 | 0.0184 | 0.0207 | 0.0277 | 0.0325 | 0.0237 | 0.0223 | 0.0238 |
| 10 | 0.0214 | 0.0219 | 0.0197 | 0.0192 | 0.0187 | 0.0210 | 0.0281 | 0.0326 | 0.0243 | 0.0225 | 0.0241 |

Figure 5 shows the component formation results of the human model on four different datasets, where different colored body parts represent different types of components. In order to further verify the robustness of the proposed method, we conduct dynamic clustering on non-human models, as shown in Figure 6. It shows that our method achieves equally effective clustering results on these non-human models.

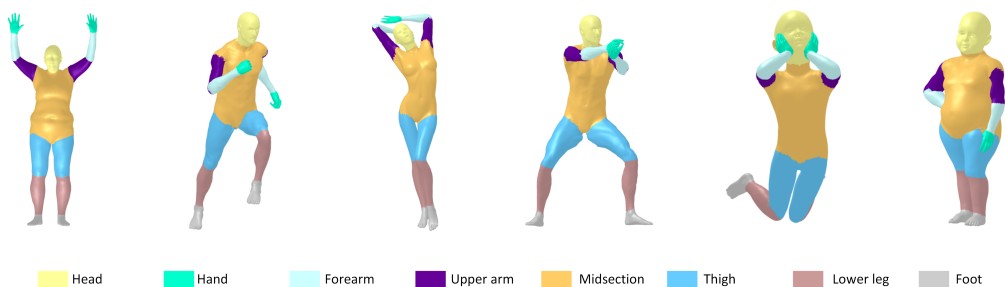

**Figure 5.** Visualization of the components of the human model in FAUST, SCAPE, TOSCA, and KIDS datasets.

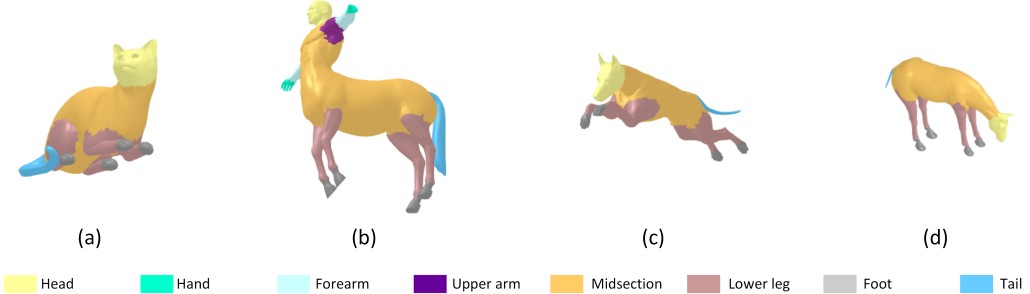

**Figure 6.** Clustering results of the proposed method on the animal model in the TOSCA dataset. (**a**) cat, (**b**) centaur, (**c**) dog, and (**d**) horse.

After the components are formed, the component of different models is matched to obtain a pair of components. Figure 7 visualizes the matching results of different models, where the left side shows the matching results of components with a different pose of the same subject, and the right side shows the matching results of components with different subjects. Experiments demonstrate that our method achieves good results in different model component matching, which proves the robustness of our method.

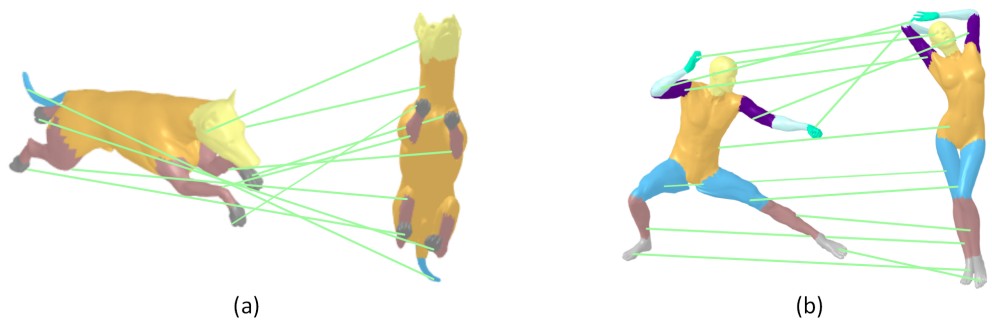

(a)                                    (b)

**Figure 7.** Component matching results of different models. (**a**) is the component matching result of the same subject with different poses, and (**b**) is the component matching result of different subjects.

*4.3. Correspondence Results*

In this section, we evaluate the experimental results of the proposed algorithm and other classical algorithms on four non-rigid datasets. We test our model on the original FAUST dataset, and Figure 8 shows the correspondence results, where intra-pair represents different poses of the same subject and inter-pair represents different subjects.

Intra-pair                                   Inter-pair

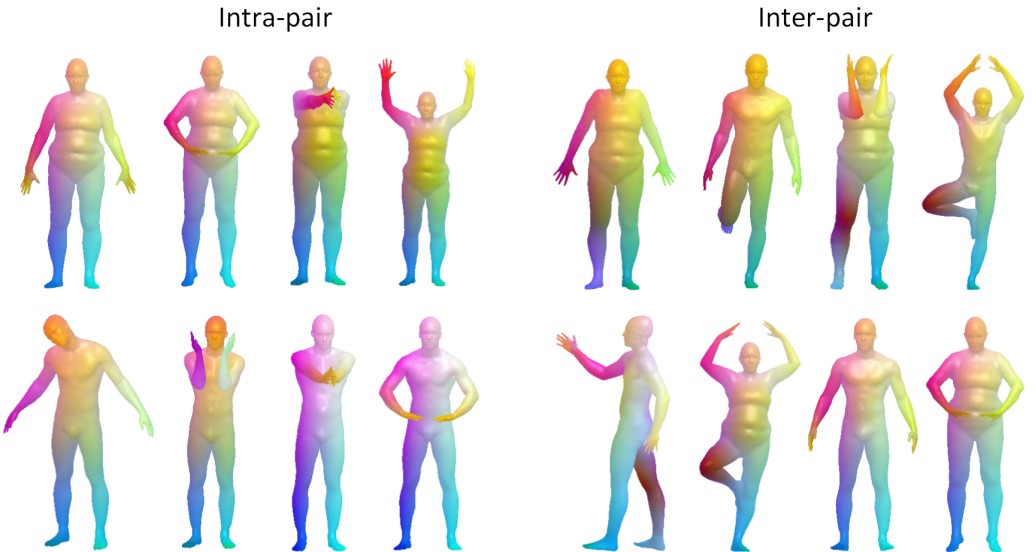

**Figure 8.** Correspondence results on the original FAUST dataset. The four columns on the left are intra-pairs, and the four columns on the right are inter-pairs. In all matched pairs, the left side is the reference shape and the right side is the matching result of our method.

The average error (AE) of our method compared with other state-of-the-art methods on the original FAUST dataset is shown in Table 3, including the average error of intra-pair and the average error of inter-pair. On the intra-pairs, our method achieves 1.85 cm, which outperforms ResNet-LDDMM significantly by 4.15%. On the inter-pairs, our method achieves 2.37 cm, which even outperforms the state-of-the-art method by 9.20%. Our method has the smallest average error in inter-pair and also performs well in intra-pair. Furthermore, SP [18] uses distributed inference to predict the model in the model initialization phase and combines local-based human pose inference with graph form human body shape models, the average error is smaller in the intra-pair. However, our method is more concerned with the generalization of shape matching, the combined average error is smaller than SP.

**Table 3.** Average errors of different methods on the original FAUST dataset. The error is expressed as the distance between the mapping point and the ground truth (cm).

| Method | Intra AE | Inter AE | Average |
|---|---|---|---|
| FMNet [12] | 2.44 | 4.83 | 3.635 |
| Cyclic-FM [21] | 2.12 | 4.07 | 3.095 |
| SP [18] | 1.57 | 3.13 | 2.350 |
| 3D-CODED [8] | 1.98 | 2.88 | 2.430 |
| FARM [22] | 2.81 | 4.12 | 3.465 |
| SURFMNet [13] | 1.73 | 3.63 | 2.680 |
| MGCN [23] | 2.51 | 3.65 | 3.080 |
| ResNet-LDDMM [24] | 1.93 | 2.61 | 2.270 |
| Ours | 1.85 | 2.37 | 2.110 |

In order to verify the robustness of the proposed method, we resample the original FAUST dataset to 4096 vertices for testing. Figure 9 visualizes the matching results of some methods on the remeshed FAUST dataset. It shows that our method achieves the best matching results in deformable shape correspondence.

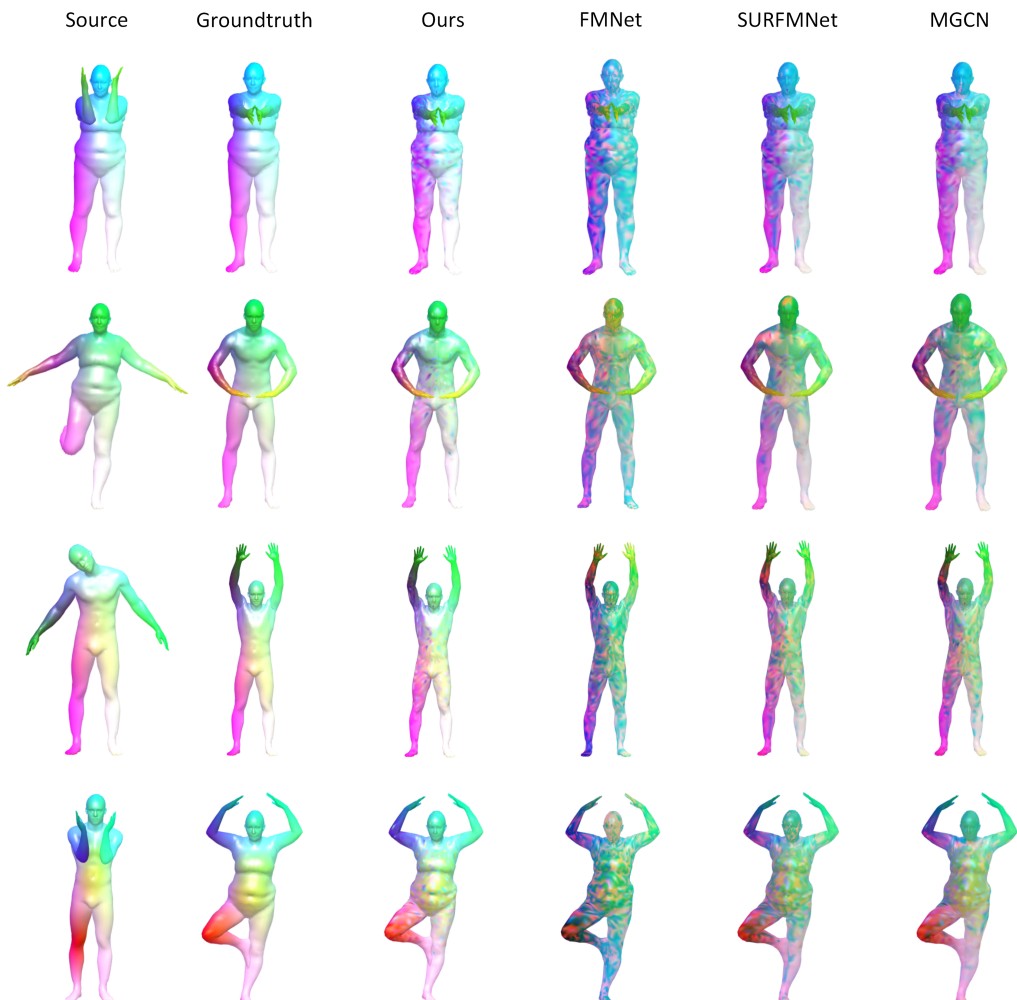

**Figure 9.** Correspondence results on the remeshed FAUST dataset. The first column shows the reference shape, the second column shows ground truth, and columns three through six show the corresponding results of different networks.

Furthermore, the correspondence performance of several methods is compared on the FAUST dataset and the remeshed FAUST dataset by using the Princeton benchmark protocol [39], as shown in Figure 10. With the increase of the geodesic error threshold, the matching accuracy is improved gradually. On the original FAUST dataset, our method is slightly improved compared with FMNet, SURFMNet, and MGCN, while the corresponding accuracy of our method is significantly improved compared with other methods on the remeshed FAUST dataset. Our approach achieves good matching for meshes of different resolutions, which indicates that it has strong robustness.

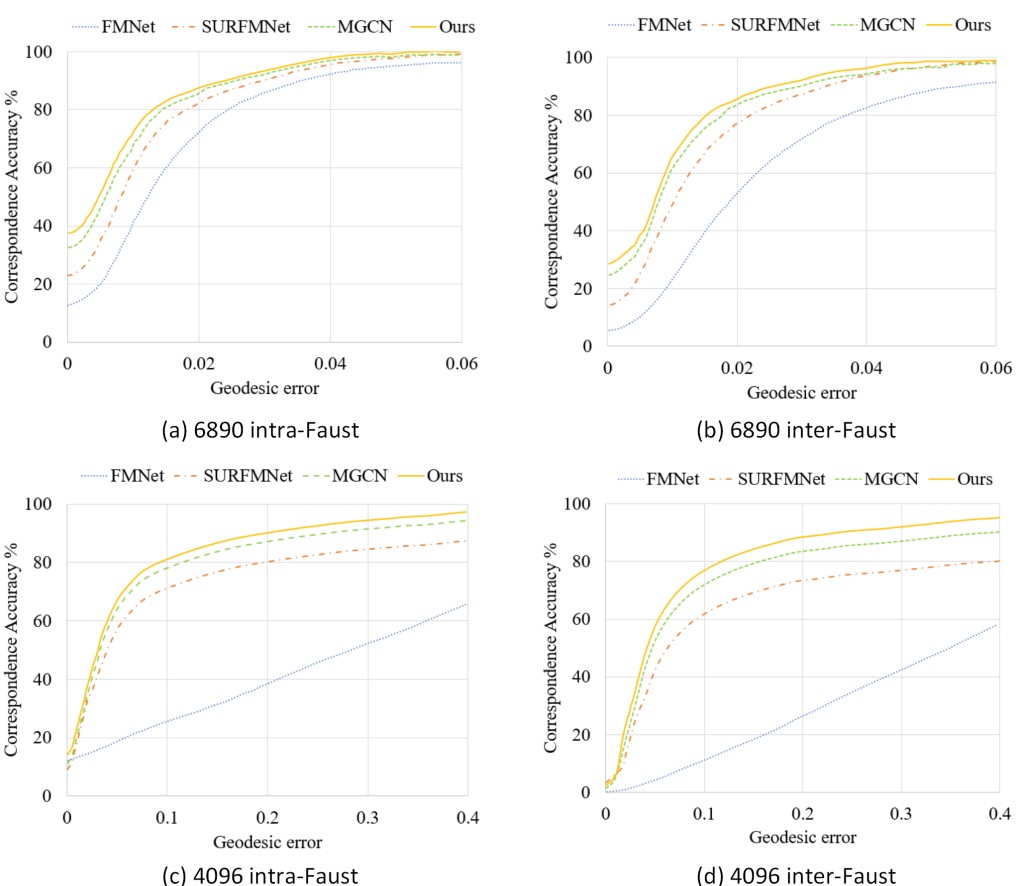

**Figure 10.** Quantitative correspondence performance of different methods on the original FAUST dataset and the remeshed FAUST dataset. (**a**,**b**) are the evaluation on the original FAUST dataset, (**c**,**d**) are the evaluation on the remeshed FAUST dateset.

We further test on SCAPE, TOSCA, and KIDS datasets to verify the generalization of our approach. For a more intuitive comparison of experimental results, Figure 11 illustrates the geodesic errors of these methods on remeshed SCAPE, TOSCA, and KIDS datasets. In order to clearly demonstrate the corresponding effect, we visualize the correspondence results of different methods on these three datasets, as shown in Figure 12. Our method shows more accurate matching results, which indicates that our method has better generalization.

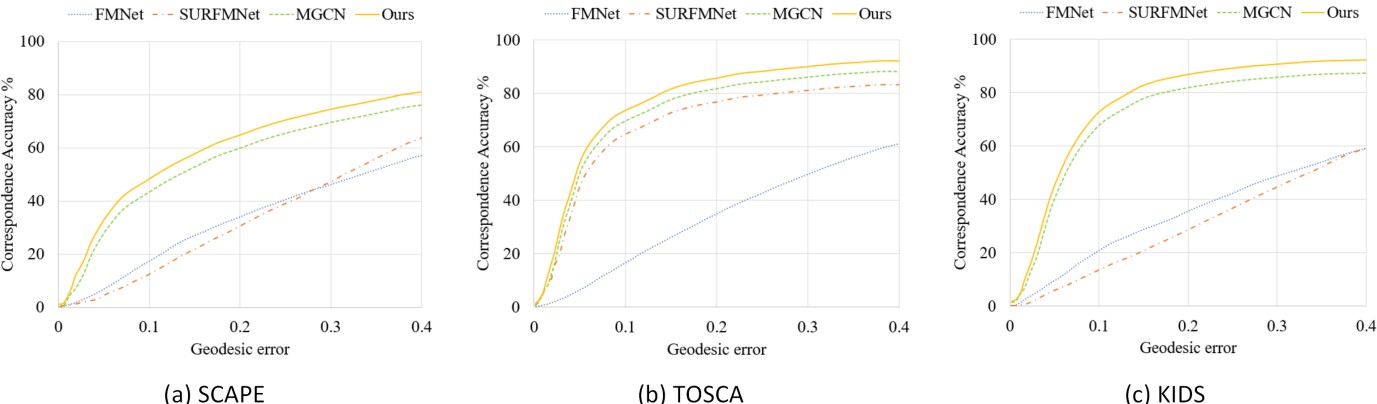

**Figure 11.** Quantitative correspondence performance of proposed methods on three remeshed datasets. (**a**) is the evaluation on the remeshed SCAPE dataset, (**b**) is the evaluation on the remeshed TOSCA dataset, (**c**) is the evaluation on the remeshed KIDS dataset.

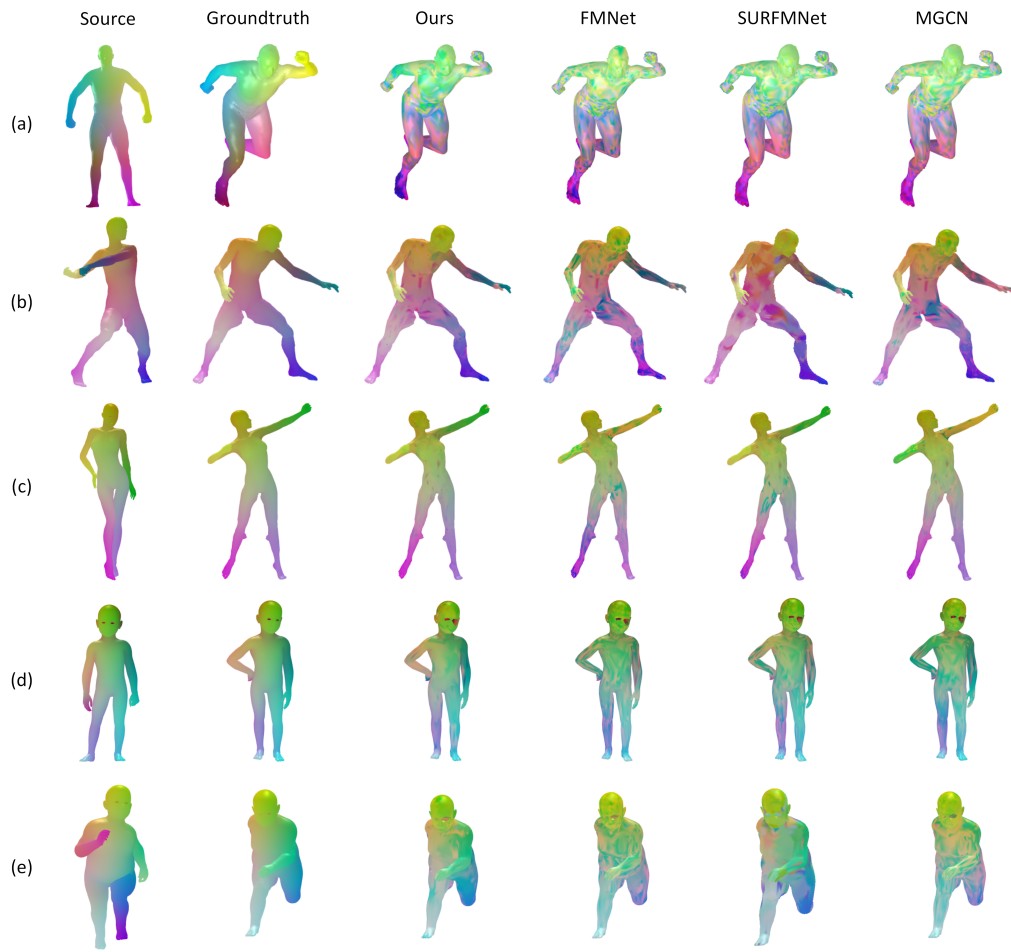

**Figure 12.** Visualization of the corresponding results of our method and other algorithms on the human model in remeshed SCAPE, TOSCA, and KIDS datasets. (**a**) SCAPE. (**b**) and (**c**) TOSCA. (**d**,**e**) KIDS. Corresponding points show the same color.

Considering the large gap between the animal model and the human model, the proposed method is tested on the animal model in the TOSCA dataset to verify the adaptability of the method. Figure 13 shows the corresponding results of reference shapes and deformable shapes on the non-human models. By contrast with FMNet, SURFMNet, and MGCN, our method is closer to ground truth in corresponding details.

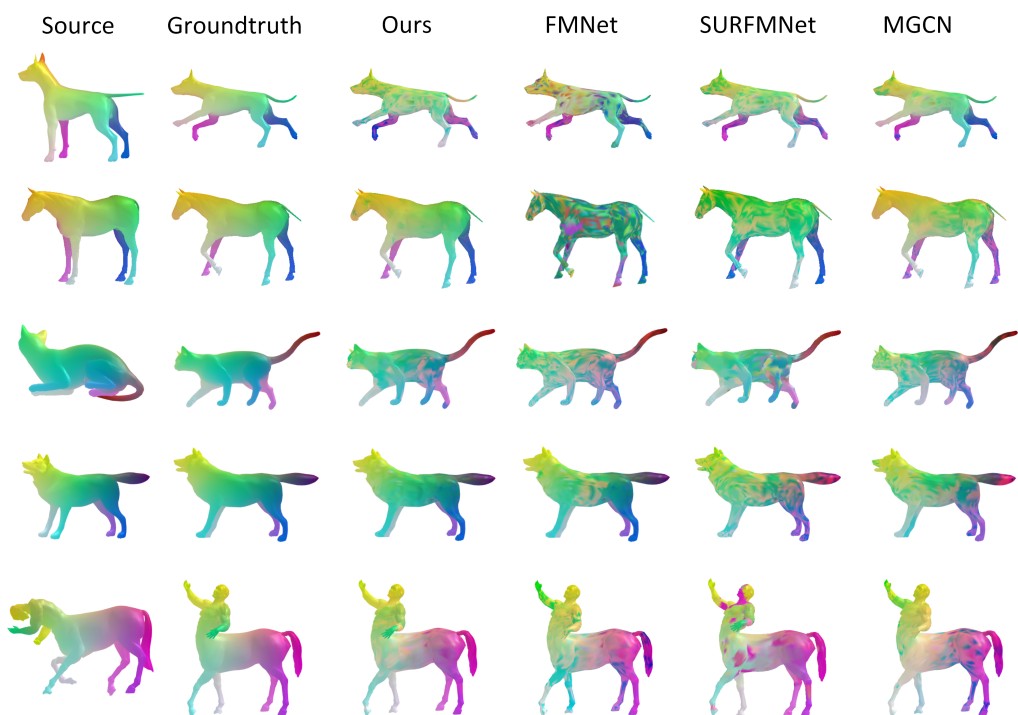

**Figure 13.** Correspondence results on the non-human model in remeshed TOSCA datasets.

In Table 4, we show the average geodesic error of several methods on four resampled datasets. The bold represents the minimum error in each column. In the remeshed FAUST dataset, the error of our method is 78.72% less than FMNet, 56.60% less than SURFMNet, and 47.43% less than MGCN. In remeshed SCAPE, it is 48.88% less than FMNet, 45.42% less than SURFMNet, and 27.71% less than MGCN. In the remeshed KIDS dataset, it is 62.32% less than FMNet, 65.66% less than SURFMNet, and 57.74% less than MGCN. In remeshed TOSCA, it is 74.65% less than FMNet, 76.16% less than SURFMNet, and 71.87% less than MGCN. It can be found that our model achieves better results than those of the compared methods, which demonstrates that our model is more robust on non-rigid 3D models with lower resolution.

**Table 4.** The average geodesic error (cm) with some methods on remeshed FAUST, SCAPE, KIDS, and TOSCA datasets. Average error is normalized to eliminate the effects of shape scale transformations.

| Method | FAUST | SCAPE | KIDS | TOSCA | | | | | | | |
|---|---|---|---|---|---|---|---|---|---|---|---|
| | | | | David | Michael | Victoria | Cat | Centaur | Dog | Horse | Wolf |
| FMNet | 0.3601 | 0.3709 | 0.3402 | 0.3413 | 0.3712 | 0.3100 | 0.3752 | 0.3574 | 0.3745 | 0.3522 | 0.0360 |
| SURFMNet | 0.3952 | 0.3895 | 0.3727 | 0.1673 | 0.3485 | 0.3572 | 0.4084 | 0.3659 | 0.3792 | 0.3496 | 0.0545 |
| MGCN | 0.3211 | 0.2755 | 0.2752 | 0.1381 | 0.2631 | 0.2787 | 0.3153 | 0.3015 | 0.3398 | 0.3045 | 0.1779 |
| Ours | 0.1357 | 0.0825 | 0.0596 | 0.0726 | 0.1902 | 0.0864 | 0.1099 | 0.0741 | 0.1158 | 0.0913 | 0.0184 |

### 4.4. Ablation Study

#### 4.4.1. Effectiveness of the Component Pair Constraint

In Section 3.2, we introduce the component pair generation strategy. The purpose of this section is to study whether component pair constraints affect network correspondence results. Figure 14 visualizes the geodesic error on the original FAUST dataset. The correspondence error on the basis of the geodesic distance to the ground truth is demonstrated with a color scale of 0 to 0.1. As shown in this figure, our model achieves a lower geodesic error on the original FAUST dataset by using a component pair constraint, which proves the effectiveness of the component pair constraint.

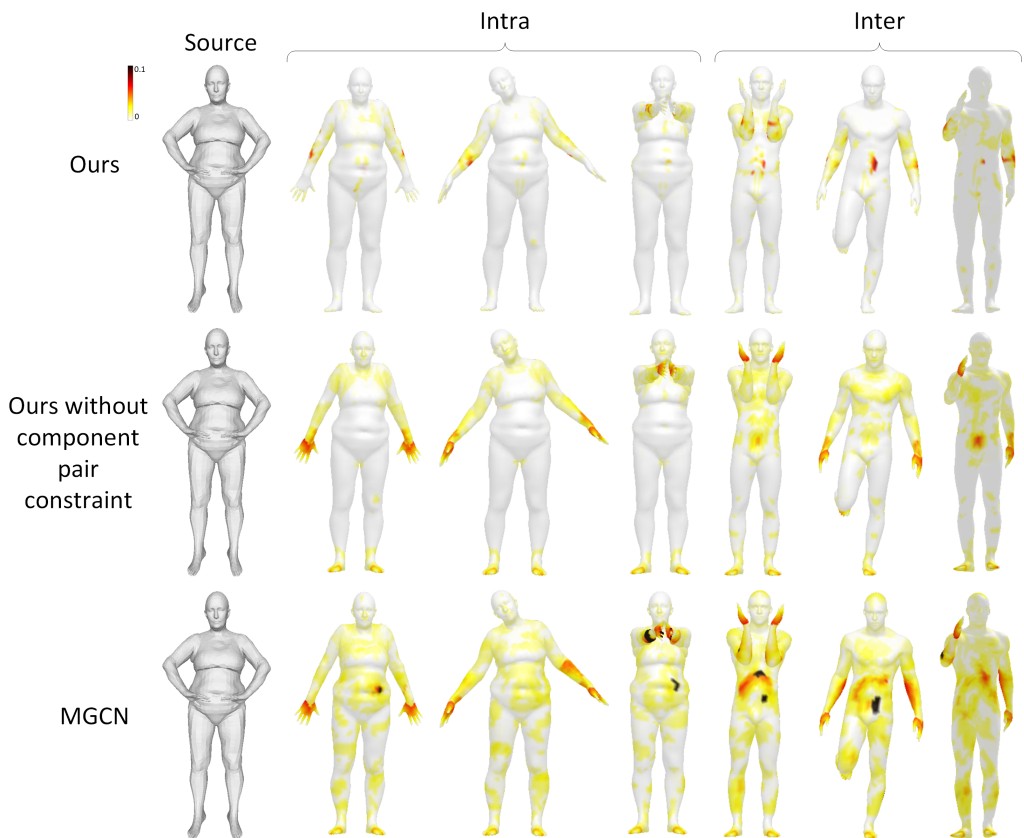

**Figure 14.** Visualization of geodesic error of several methods on original FAUST dataset.

4.4.2. Effectiveness of the Component-Aware Graph Routing

In Section 3.3, component-aware graph routing is proposed to compute the primary capsule to transfer the input vector to the latent capsule. To test the effectiveness of the routing algorithm, we compare the proposed algorithm with dynamic routing. Table 5 shows the average geodesic error of different routing algorithms on the TOSCA dataset. The bold represents the minimum error in each column. As shown in Table 5, the average geodesic error of the dynamic routing algorithm is 0.1276, and the average geodesic error of our method without knowledge distillation is 0.1213. Our method improves the correspondence performance with an average error of 0.1187, which is 6.97% less than the dynamic routing and 2.14% less than our method without knowledge distillation. Our model achieves smaller average geodesic errors on different types of meshes, providing slightly better results on the TOSCA dataset. It illustrates that the routing strategy in this paper can capture more adequate pose information compared to dynamic routing, resulting in significant improvement in shape correspondence.

**Table 5.** Comparison of average geodesic error (cm) on remeshed TOSCA dataset using different routing strategy.

| Datasets | Dynamic Routing | Ours without Knowledge Distillation | Ours |
|---|---|---|---|
| david | 0.1153 | 0.1193 | 0.1186 |
| michael | 0.1295 | 0.1136 | 0.1129 |
| victoria | 0.0644 | 0.0612 | 0.0598 |
| cat | 0.1058 | 0.0949 | 0.0941 |
| centaur | 0.1824 | 0.1698 | 0.1605 |
| dog | 0.1625 | 0.1579 | 0.1572 |
| horse | 0.1137 | 0.1182 | 0.1174 |
| wolf | 0.1471 | 0.1354 | 0.1287 |

In order to fully capture the features of the potential space, inter-layer distillation and output-layer distillation are used to transfer the model's knowledge. To verify the effectiveness of the knowledge distillation strategy, we train the CA-GCNet and the CA-GCNet without a knowledge distillation strategy separately to acquire the network accuracy, as shown in Figure 15. The accuracy curves illustrate that the prediction accuracy of our proposed CA-CGNet is higher than that of the CA-CGNet without the knowledge distillation strategy, which indicates that the student network can learn more detailed features from the teacher network.

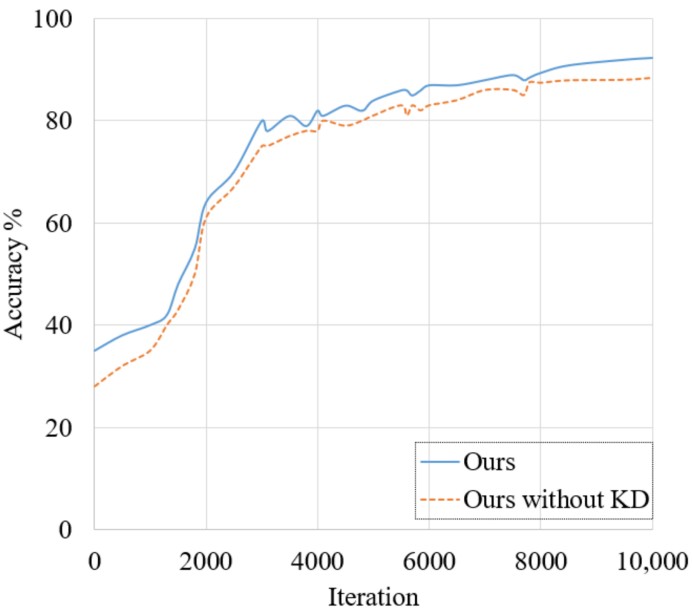

**Figure 15.** The accuracy of our CA-CGNet and CA-CGNet without knowledge distillation in 10,000 iterations.

## 5. Discussion

As shown in Figure 14, the functional maps with component pair constraint can capture the local pose details of 3D non-rigid models more effectively. Hence, functional maps with component pair constraints are able to use deep features with local and global structural information for shape correspondence. To reduce the effect of noise interference, the dynamic clustering strategy is used to dynamically merge the patches formed by over-segmentation, capturing the potential mutual influence of information between the local patch and the global model. Then, component pairs are obtained by matching components on different models using the Hausdorff distance, which leads to high-level feature representation and corresponding accuracy by using component pair-constrained functional maps.

Inter-layer distillation and output-layer distillation are designed to constrain the student network to reduce the network parameters and speed up the gradient descent during the training process. The average geodesic error shown in Table 5 illustrates the efficiency of our component-aware graph routing for improving the corresponding accuracy of 3D models with low resolution.

## 6. Conclusions

Based on the functional maps framework, this paper proposes a component-aware shape correspondence network that can effectively deal with non-rigid deformation. Taking capsules as nodes in the graph, the semantic information between capsules is further extracted by the component-aware graph routing, and the model is simplified by a knowledge distillation strategy. By adding the component constraint to the functional map and taking the component-based semantic loss as a regularization term, our method can learn

more representative features on 3D meshes. Experiments on four challenging datasets show qualitatively and quantitatively that our model has stronger robustness and better generalization ability. The proposed framework also can provide more effective solutions in biological computing, human pose estimation, and medical image processing. One limitation of dynamic clustering is that it is difficult to cluster patches if the feature difference is hardly remarkable, which we hope to address in a future study. On the other hand, the determination of the number of clusters requires a large number of iterative experiments and is computationally intensive. In future work, we will adopt an optimization algorithm to determine the number of clusters to reduce computational expenditure.

**Author Contributions:** Conceptualization, Y.L. and M.C.; methodology, Y.L.; software, M.C.; validation, Y.L.; formal analysis, M.C.; investigation, Y.L.; resources, M.C.; data curation, Y.L.; writing—original draft preparation, M.C.; writing—review and editing, Y.L.; visualization, M.C.; supervision, Y.L.; project administration, Y.L.; funding acquisition, Y.L. All authors have read and agreed to the published version of the manuscript.

**Funding:** This research was funded by NSFC 61972353, NSF IIS-1816511, OAC-1910469 and Strategic Cooperation Technology Projects of CNPC and CUPB: ZLZX2020-05.

**Institutional Review Board Statement:** Not applicable.

**Informed Consent Statement:** Not applicable.

**Data Availability Statement:** Not applicable.

**Conflicts of Interest:** The authors declare no conflict of interest.

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
