# Peer review of "CA-CGNet: Component-Aware Capsule Graph Neural Network for Non-Rigid Shape Correspondence"

_applsci, doi:10.3390/app13053261_

Round 1

Reviewer 1 Report

The authors present a method to perform Non-rigid Shape Correspondence based on modifications and combinations of existing methods in a unique way. There methodology seems to produce quality results, however there are some points that need clarification.

1. in section 2.2 the second sentence needs to be rewritten as it is difficult to follow the thoughts.

2. The dynamic clustering needs to be explained in conceptual terms as well as mathematics. The methodology sounds very similar to hierarchical approaches and it is not clear why this methodology should be used over others.

3. Multiple different distance metrics are used at different points including Hausdorff and Wasserstein.  Why were these used over others, why not the same distance metric at every point?

4. Table 1 shows the identification error, but no scale or units are provided to give them meaning.

5. Figures 4 and 5 showing cluster results, but it is unclear what I am looking for in these figures that indicates success. also, what do the colors represent. This is a question for all similar figures.

6. All figures rely heavily on colors that do not show well if printed in black and white.  If possible please alter the visualization to allow for clarity in black and white.

7. Tables 3 and 4 are also unclear in numbers, what is a good number?

Reviewer 2 Report

The authors of this paper propose a Component-aware Capsule Graph Network to address the characteristics of embedding space based on component constraints. To reduce noise interference, a dynamic clustering strategy is used to classify the features of patches produced by over-segmentation. In addition, the authors proposed a component-aware capsule graph routing method to fully describe the relationship between capsules, which treats capsules as nodes in a graph network and constraints nodes via component information. Then, a knowledge distillation strategy is introduced to improve the network's convergence speed by reducing parameters while maintaining accuracy. A component pair constraint is added to the functional map, and a component-based semantic loss function that can compute isomeric functions in both direct and symmetric directions is proposed. My main concerns are the lack of related work, the approach presentation, and the experimental section. The authors do not mention any related work, but they do compare their approach to other approaches in the literature. I would recommend that authors provide a clear description of their contribution to the other approaches.

Section 1, The authors cite some works in the Introduction section's references. They have not, however, indicated the advantage or disadvantage and their relationship to this paper. It's a little perplexing.

Section 2, The author's proposal to overcome the issues in the current model should be included in the literature review techniques. Authors must strengthen the literature review to give readers an understanding of the research positioning in the body of knowledge.

Section 3, If there is any new methodology, it has not been properly highlighted. Furthermore, the paper structure and the proposed work should be illustrated well  

Authors should provide a detailed figure to describe the proposed solution so that the reader can quickly understand the method.

The mapping process for the proposed model should be discussed in detail. (more details needed!)

Section 4, Clarifies the finding Error rate and accuracy in the performance analysis. The experiments’ analysis should be enhanced. For example, the authors should give a more detailed analysis of the Tables in this section.

The authors should add more details about the implementation of the code to perform the analysis and the library involved in this task. ( add the link of the code to check it)

authors proved that their model can generate more acceptable, but have not indicated the practical application and effects of such a solution

A comparison with recent research and methods would be helpful. ( plz add 1-2 recent models plus in table 2)

Section 5, The authors must discuss their theoretical contributions to the proposed model in comparison to other models in the Conclusion section. This should be done in a separate paragraph.

Finally, There are numerous typos in this article. The authors must correct it. Language editing is advised.

Reviewer 3 Report

The authors have designed a component-aware capsule graph network (CA-CGNet) for the non-rigid shape correspondence. They have employed the component-aware capsule graph routing to find the relationship between capsules. They have achieved good results, which are shown both qualitatively and quantitatively. This work could be used for various real-world applications. Besides this, I would give the authors a few suggestions and ask for a few corrections, which are as follow:

1. There are a few spelling mistakes and typos, which should be removed such as, in lines, 32,33  "scholars have proposed further to"  and in line 274, the word wights is replaced with weights.

2. If the knowledge distillation strategy increases the convergence speed and decreases the number of parameters; why then it helps to improve the accuracy, shouldn't it be the other way around?

3. Please look into equation number 5. 

4. There are various coefficients that are part of some of the equations like; lambda in eq (6) and mu1 and mu2 in eq (16), which may give not favorable outcomes. What are the values of these coefficients, if possible write them down in a single line, or if that varies for various cases then adding a table would be good? 

5. The datasets you have used are the 3D models with less number of vertices, can you please run your algorithm on the point clouds with a large number of vertices as the graph construction on the small 3D models is easy and less computationally expensive? 

6. You have tested the different numbers of clustering and then with the help of identification errors, you have selected the k for human and non-human subjects in the datasets. However, this seems computationally costly and a hit-and-trial method. What do you think, if one uses the optimization algorithm for the appropriate selection of k in clustering?

Round 2

Reviewer 1 Report

Thank you for addressing my comments and concerns